# Effects of Gallic Acid on In Vitro Ruminal Fermentation, Methane Emission, Microbial Composition, and Metabolic Functions

**DOI:** 10.3390/ani15131959

**Published:** 2025-07-03

**Authors:** Wei Zhu, Jianjun Guo, Xin Li, Yan Li, Lianjie Song, Yunfei Li, Baoshan Feng, Xingnan Bao, Jianguo Li, Yanxia Gao, Hongjian Xu

**Affiliations:** 1College of Animal Science and Technology, Hebei Agricultural University, Baoding 071001, China; 18033640837@163.com (W.Z.); 15039055246@163.com (Y.L.); ashan0711@163.com (B.F.); 17835713549@163.com (X.B.); jgli@hebau.edu.cn (J.L.); 2Chengde Academy of Agriculture and Forestry Sciences, Chengde 067000, China; gjj730209@126.com (J.G.); songlianjie1993@163.com (L.S.); 3College of Veterinary Medicine, Hebei Agricultural University, Baoding 071001, China; m13503666661@163.com (X.L.); dyly@hebau.edu.cn (Y.L.); 4Key Laboratory of Healthy Breeding in Dairy Cattle (Co-Construction by Ministry and Province), Ministry of Agriculture and Rural Affairs, Baoding 071000, China

**Keywords:** gallic acid, methane emission, rumen fermentation, metagenomics

## Abstract

The greenhouse effect is an urgent global environmental issue. Enteric methane from ruminant livestock production accounts for about 40% of livestock’s greenhouse gas emissions. Reducing enteric methane emissions is crucial for the sustainability of ruminant production; one of the most effective strategies for this is the use of natural feed additives. Gallic acid, a phenolic compound and monomer of hydrolysable tannin with antimicrobial properties, has the potential to mitigate methane emissions. Therefore, this study aimed to assess the impact of gallic acid on nutrient digestibility, methanogenesis, rumen fermentation, and the microbial community and its functions in dairy cows using in vitro fermentation techniques. In this study, the inclusion of GA in the diet modified microbial community and functions, leading to altered rumen fermentation and reduced methane emissions without affecting fiber degradability. An optimal GA feeding rate in the present in vitro study appears to be 10 mg/g DM.

## 1. Introduction

Greenhouse gas emissions significantly contribute to environmental issues such as climate change and global warming. Methane accounts for 20% of total greenhouse gases and is the second largest contributor to global warming, with its impact being 25 times greater than that of CO_2_ [1,2]. The Intergovernmental Panel on Climate Change reported that livestock emissions account for 6.3% of anthropogenic greenhouse gas emissions, with enteric methane from ruminants contributing 35–40% of these emissions [3]. Furthermore, methane emission as a byproduct of rumen microbial fermentation results in a direct energy loss of 2 to 12% for ruminants [4]. Therefore, the environmental impact and energetic inefficiency of ruminal methane emissions have attracted increasing attention, and scholars have carried out intensive research to develop strategies to mitigate methane emissions or modulate ruminal fermentation. However, these strategies have shown varying degrees of success, with some facing numerous challenges and limitations in practical farm applications, such as negative impacts on animal health and productivity.

Reducing enteric methane emissions is important for the sustainability of ruminant production. Enteric methane mitigation can be achieved by either directly inhibiting methanogens to reduce hydrogen production or by alternative rumen metabolism that utilizes hydrogen [5]. Numerous studies suggested that enteric methane mitigation can be inhibited by plant secondary metabolites, including phenolic compounds and their functional derivatives, owing to their antimicrobial properties [6,7]. Meanwhile, the utilization of phenolic compounds by rumen microorganisms could represent a pathway for H_2_ transformation into beneficial products, potentially improving production efficiency [8]. For instance, specific rumen bacteria can utilize H_2_ to catabolize phenolic compounds such as gallate and phloroglucinol, producing acetate as an energy-yielding product for animals [9]. Gallic acid, a phenolic compound or monomer of hydrolysable tannin, has been shown to decrease H_2_ accumulation and archaeal abundance while increasing acetate and butyrate production in in vitro rumen fermentation [10]. Furthermore, tannins and their bioactive compounds, which are polyphenolic, have been confirmed as alternatives to antibiotics in animal nutrition, modulating ruminal fermentation, amino acid deamination, energetic inefficiency, and methane emissions by altering microbial populations. Tannic acid, a hydrolysable tannin, has also been reported to interact with rumen microbes, affecting digestibility and rumen fermentation. A low dietary concentration has an inhibitory effect on enteric methane and improves N utilization efficiency [11,12].

In view of the roles of gallic acid and its metabolites as both a regulator of rumen microbiomes and an H_2_ acceptor that captures excess hydrogen to produce useful end products, it may mitigate methane production and energy loss. However, the mechanisms by which gallic acid affects the rumen microbial community and function in dairy cows remain unclear. Therefore, the objectives of this study were to investigate the effects of gallic acid on ruminal in vitro nutrient digestibility, methane production, fermentation parameters, and the rumen microbial community and its functions.

## 2. Materials and Methods

The experiments were conducted at the College of Animal Science and Technology, Hebei Agricultural University (Baoding, China), and animal procedures were approved by the Experimental Animal Ethics Committee of Hebei Agricultural University (2023052).

### 2.1. Experimental Design, Animals, and Diet

A completely randomized design was used for the in vitro incubation. The experiment included three 24 h in vitro cultures, with each culture comprising 36 samples: 5 treatments with 6 replicates each, plus 6 blanks containing only inoculum to account for background gas production. The 5 treatments comprised different gallic acid dose levels (0, 5, 10, 20, and 40 mg/g DM), defined as control, GA1, GA2, GA3, and GA4, respectively. The selection of GA doses was based on those used in previous experiments [13,14]. The gallic acid used (purity > 90%) was supplied by Kemin Industries Co., Ltd. (Ningbo, Zhejiang, China). Rumen fluid was collected from three fistulated multiparous lactating Holstein cows that received a total mixed ration (TMR) comprising the following (on a DM basis): corn silage (31.7%), alfalfa hay (17.8%), soybean meal (11.1%), rapeseed meal (2.3%), cottonseed meal (2.2%), steam flaked corn (18.7%), dried distillers’ grains with solubles (4.9%), wheat bran (9.6%), and a minerals and vitamins premix (1.7%).

### 2.2. In Vitro Incubation

Rumen fluid was collected before morning feeding from three donors, mixed in equal volume, and placed in a pre-warmed thermos. The rumen fluid was filtered through 4 layers of gauze The artificial buffer solution was prepared as described by Liu et al. [2], with each liter containing 8.75 g of NaHCO_3_, 1.00 g of NH_4_HCO_3_, 1.43 g of Na_2_HPO_4_, 1.55 g of KH_2_PO_4_, 0.15 g of MgSO_4_·7H_2_O, 0.52 g of Na_2_S, 0.017 g of CaCl_2_·2H_2_O, 0.015 g of MnCl_2_·4H_2_O, 0.002 g of CoCl·6H_2_O, 0.012 g of FeCl_3_·6H_2_O, and 1.25 mg of resazurin. The rumen inoculum was then prepared by mixing the rumen filtrate and artificial buffer solution at a ratio of 1:2 (*v*/*v*), followed by continuous CO_2_ flushing at 39 °C to maintain an anaerobic environment. The substrate was identical to the TMR fed to the donor cows. It was dried at 55 °C for 48 h and then passed through a 1 mm sieve using a Wiley mill. The dried substrate was accurately weighed at 3 g, and 450 mL of rumen inoculum was added to fermentation flasks (500 mL), which were subsequently sealed. All operations were conducted under anaerobic conditions with continuous CO_2_ flushing. The Artificial Rumen Simulation System was employed in this in vitro experiment (MC-ABSF-II, Beijing Manchang Technology Co., Ltd., (Beijing, China). The fermentation flasks were incubated at 39 °C for 24 h with horizontal shaking at 60 rpm. After 24 h, the incubation was halted, and the fermentation flasks were submerged in ice water to stop the fermentation. Preweighed nylon bags (42 μm) were used to filter the entire biomass material of each flask. The samples were collected and preserved for subsequent analysis.

### 2.3. Sample Collection and Analysis

In vitro gas production was performed using a fully automated technique in the Artificial Rumen Simulation System (MC-ABSF-II, Beijing Manchang Technology Co., Ltd.) that recorded and corrected the total gas volume for standard atmospheric pressure (101.3 kPa). The released gas was automatically directed to a gas chromatograph equipped with a thermal conductivity detector for measuring CH_4_ and CO_2_ concentrations. The residues in nylon bags were dried at 55 °C for 48 h and then ground through a 1 mm sieve in a Wiley mill before analyzing the apparent disappearance of nutrients. The residues from nylon bags were analyzed for the content of DM (method 930.15) and CP (method 976.05) according to AOAC [15]. The pH of the incubated rumen fluid was measured using a calibrated pH meter (PHS-25, Shanghai INASE Scientific Instrument Co., Ltd., Shanghai, China). The rumen fluid was acidified by adding 2 mL 25% metaphosphoric acid per 10 mL rumen fluid before analyzing VFA and ammonia-N (NH_3_) levels. The concentrations of ammonia-N were determined using the phenol–hypochlorite method [16]. The sample was analyzed for VFAs using a gas chromatograph (7890A, Agilent Technologies Co., Ltd., Santa Clara, CA, USA). The content of microbial protein (MCP) was determined following the procedures outlined by Hu et al. [17]. Samples for microbial DNA extraction and metagenome sequencing were immediately frozen in liquid nitrogen and then stored at −80 °C (Table 1).

### 2.4. DNA Extraction and Metagenome Sequencing

Total genomic DNA from rumen fluid samples was extracted using a repeated bead-beating plus column method [18]. The DNA integrity was assessed by agarose gel electrophoresis, and quality and concentration were analyzed by ultraviolet spectrophotometer (Thermo Fisher Scientific, Wilmington, DE, USA). A paired-end library was constructed using TruSeq Nano DNA Library Preparation Kit following the manufacturer’s instructions (FC-121-4001, Illumina, San Diego, CA, USA), and paired-end sequencing was performed on HiSeq4000 (150 bp paired-end). The 3′ and 5′ ends of the paired-end reads were removed with SeqPrep, and low-quality reads (length < 50 bp, quality value < 20, or N bases) were removed using Sickle (version 1.33). Quality-filtered reads were first aligned to the bovine genome (bosTau8 3.7) using BWA (Version 0.7.9a) to filter out host contamination [19]. The remaining reads were assembled individually for each sample using Megahit (Version 1.2.9) [20]. Open reading frames (ORFs) were predicted from assembled contigs longer than 500 bp using MetaGene. Assembled contigs were clustered with CD-HIT, and non-redundant sequences were identified from gene sets with at least 95% identity [21]. Original sequencing reads were mapped to predicted genes to estimate their abundances using SOAPaligner (Version 2.2.1) [22].

The α-diversity index was calculated from the relative abundance data of microbial species using R software (version 4.3.2). The PCoA based on Bray–Curtis dissimilarity matrices was conducted to visualize species-level taxonomic composition. The BLASTP (version 2.2.28+) was used for taxonomic profiles [23] by aligning the non-redundant gene catalogues with the NCBI NR database [24]. Taxonomic profiles were summarized at the phylum, genus, and species levels by calculating the relative abundances for each rank. Microbial taxa with a relative abundance greater than 0.1% in at least 50% of samples per group were selected for further analysis. KEGG pathways were annotated using Diamond against the KEGG database. The abundance of KEGG pathways was normalized to counts per million reads (cpm) for further analysis.

### 2.5. Statistical Analysis

All data were checked for normality and outliers using the UNIVARIATE procedure in SAS (version 9.4, SAS Institute Inc., Cary, NC, USA) before analyzing the data. The gas production, nutrient digestibility, and rumen fermentation parameters were statistically analyzed using the PROC MIXED procedure in SAS. The statistical model used was Yij = μ + Ri + Tj + eij, where Yij = dependent variable, μ = overall mean, Ri = run (i = 1, 2, 3), Tj = treatment (j = 1, 2, 3, 4, 5), and eij = random error. Treatment effects were considered fixed, and replication was considered a random effect. Polynomial contrasts were used to test the linear, quadratic, and cubic effects of treatments (the inclusion rate of gallic acid). The PDIFF statement was used to separate differences among treatments. Differences were declared significant at *p* ≤ 0.05, and a trend was noted at 0.05 < *p* ≤ 0.10.

## 3. Results

### 3.1. Nutrient Degradability and Gas Production

The digestibility of DM, crude protein (CP), neutral detergent fiber (NDF), and acid detergent fiber (ADF), as well as the production of total gas, CH_4_, and CO_2_, are presented in Table 2. The digestibility of DM and CP decreased quadratically (*p* < 0.05) with increasing gallic acid doses. The group with a 10 mg/g DM dose of gallic acid exhibited lower DM and CP digestibility compared to other groups. The total gas production showed a quartic response (*p* = 0.05), decreasing to the lowest level in the 10 mg/g DM dose group, then increasing with further gallic acid inclusion. The CH_4_ production and CH_4_/total gas were quadratically decreased (*p* < 0.0001) with increasing gallic acid dose. Increasing gallic acid inclusion quadratically decreased CO_2_ production (*p* = 0.001) and CO_2_/total gas (*p* = 0.006). The CH_4_ and CO_2_ production, as well as their ratios to total gas, were the lowest in the 10 mg/g DM dose group.

### 3.2. In Vitro Fermentation Parameters

In vitro ruminal fermentation pH, NH_3_-N, MCP, and VFA production of diets supplemented with gallic acid are presented in Table 3. Ruminal pH showed a tendency toward a linear reduction with increasing gallic acid doses (*p* = 0.06). The NH_3_-N concentration decreased linearly with increasing gallic acid (*p* = 0.01). The MCP concentration increased linearly (*p* = 0.0003) as gallic acid increased. Total VFA (*p* = 0.004), acetate (*p* = 0.03), and valerate (*p* = 0.03) decreased quadratically, while butyrate (*p* = 0.0006) increased quadratically with increasing gallic acid doses. The 10 mg/g DM dose group exhibited the lowest levels of total VFA, acetate, and valerate and the highest level of butyrate compared to other groups. The propionate (*p* = 0.03) and ratio of acetate to propionate (*p* = 0.03) decreased linearly with increasing gallic acid doses, and isobutyrate (*p* = 0.09) and isovalerate (*p* = 0.06) tended to decrease linearly with increasing gallic acid doses.

### 3.3. Alpha Diversity Indices and Principal Coordinate Analysis of Microbial Community

The α-diversity indices of rumen bacteria, archaea, protozoa, and fungi are shown in Table 4. The observed species of rumen bacteria in the gallic acid group tended to be higher than in the control group (*p* = 0.06). The Shannon and Simpson indices of rumen bacteria in the gallic acid group were significantly lower than the control (*p* < 0.01). The α-diversity indices of rumen archaea were not affected by gallic acid. The observed species of rumen protozoa were higher (*p* = 0.02) for the gallic acid group than for the control. The Simpson index tended to be higher (*p* = 0.054), while the Shannon index was significantly lower (*p* = 0.009) for the gallic acid group than for the control in rumen fungi.

The community structures of rumen bacteria, archaea, protozoa, and fungi, as assessed by principal coordinate analysis (PCoA) between control and GA, are shown in Figure 1. The community structures of rumen bacteria (adonis test *p* = 0.005; Figure 1a) and fungi (adonis test *p* = 0.003; Figure 1d) were significantly influenced by gallic acid. The community structures of rumen archaea and protozoa showed no significant differences between the control and gallic acid groups (adonis test *p* > 0.1; Figure 1b,c).

### 3.4. Differences in Bacterial Community

The bacterial community was explored at the phylum, genus, and species levels (Figure 2). At the phylum level (Figure 2A), compared with the control group, the relative abundances of *Bacteroidota* and *Pseudomonadota* were significantly lower (*p* < 0.05), and *Actinomycetota* tended to be lower in the GA group. The relative abundances of *Planctomycetota*, *Candidatus_Saccharibacteria*, *Chloroflexota*, *Thermodesulfobacteriota*, *Synergistota*, and *Verrucomicrobiota* were significantly higher (*p* < 0.05) in the GA group. At the genus level (Figure 2B), GA significantly reduced the relative abundances of *Bacteroides* and *Alistipes* (*p* < 0.05) and tended to decrease the relative abundances of *Bacteroidales_unclassified* (*p* = 0.065) and *Lactimicrobium* (*p* = 0.093), whereas it significantly increased the relative abundances of *Butyrivibrio* and *Oscillospiraceae_unclassified* (*p* < 0.05) and tended to increase *Eubacteriales_unclassified* (*p* = 0.065). At the species level (Figure 2C), GA supplementation significantly affected (*p* < 0.05) a total of 38 bacterial species. Among these, 29 species, including *Prevotellasp.E15-22*, *bacteriumP3*, and *Alistipessp.CAG:435*, were less abundant in the GA group, while 9 species, including *Aristaeella_lactis* and *Aristaeella_hokkaidonensis*, were significantly more abundant.

### 3.5. Differences in Archaeal Community

The archaeal community was explored at the phylum, genus, and species levels (Figure 3). At the phylum level (Figure 3A), the relative abundance of *Candidatus_Heimdallarchaeota* was significantly lower in the GA group than in the control (*p* < 0.05). At the genus level (Figure 3B), the relative abundance of *Candidatus_Heimdallarchaeota_unclassified* was significantly lower in the GA group than in the control (*p* < 0.05). At the species level, GA supplementation significantly decreased (*p* < 0.05) the relative abundances of *Methanobrevibacter_thaueri*, *Methanobrevibacter_boviskoreani*, *Methanobrevibactersp.AbM4*, and *Candidatus_Heimdallarchaeota_archaeon*.

### 3.6. Differences in KEGG Functions of Microbial Community

The KEGG pathway classification results showed that rumen microbial differential genes were predominantly enriched in global and overview maps, carbohydrate metabolism, amino acid metabolism, metabolism of cofactors and vitamins, nucleotide metabolism, glycan biosynthesis and metabolism, and energy metabolism (Figure 4A). At the second level of KEGG pathways, compared with the control group, the categories of Carbohydrate metabolism, Amino acid metabolism, and Nucleotide metabolismwere significantly enriched by the addition of GA (*p* < 0.05, Figure 4B). At the KEGG pathway definition level (level 3, Figure 4C), within the carbohydrate metabolism category, pathways such as Amino sugar and nucleotide sugar metabolism, Starch and sucrose metabolism, Glycolysis/Gluconeogenesis, and Pyruvate metabolismwere significantly enriched in the GA group (*p* < 0.05). The Alanine, aspartate and glutamate metabolism and Pyruvate metabolismwithin the Amino acid metabolism category were significantly enriched in the GA group (*p* < 0.05).

## 4. Discussion

### 4.1. Nutrient Digestibility and Gas Production

In the previous literature, the effects of tannic acids on nutrient digestibility were mixed, primarily depending on the type and dose of tannins. Bhatta et al. [25] and Animut et al. [26] reported that supplementation of tannin extracts or tannin-containing plants at concentrations above 5 g of condensed tannin per kg of DM can reduce nutrient digestibility both in vitro and in vivo. High concentrations of tannins adversely affect nutrient utilization and productivity of animals [27]. However, hydrolysable tannins significantly inhibited methanogen activity through their powerful protein precipitation ability and the antimicrobial toxicity of their degradation products; therefore, they were more effective than condensed tannins in reducing methane emission and had less negative impact on feed organic matter degradability [28]. In this study, increasing gallic acid showed a quadratic decrease in DM and CP degradation. The decrease in DM degradation was probably due to reduced CP degradation. The decreased CP digestibility by gallic acid might be attributed to the formation of tannin–protein complexes, which hinder ruminal microbial degradation of CP. This is beneficial for reducing ruminal protein degradation and increasing the availability of rumen undegradable protein. Undigested protein in the rumen will be absorbed in the small intestine. Moderate to high concentrations of tannins can exert antimicrobial effects on ruminal microbes, thereby affecting rumen fermentation [29].

As reported, hydrolysable tannins can be degraded by rumen microbes, and their gradual dissolution in the rumen can potentially influence methanogenesis, forage degradation, and microbial colonization [30]. A moderate dose of gallic acid decreased the methane production and methane/total gas in this study. Previous studies also reported that plants rich in hydrolysable tannins inhibited methane production, suggesting that hydrolysable tannins could be more effective than condensed tannins in reducing methane emissions [12,31]. Other studies showed that hydrolysable tannin extracts are more effective in reducing methane emission than condensed tannin extracts without compromising rumen fermentation and nutrient digestibility [28,32]. Additionally, gallic acid is considered an effective H_2_ acceptor, as ruminal microbiota can convert gallic acid to phloroglucinol or resorcinol, which can then be reduced to dihydrophloroglucinol using H_2_ as electron donors, thereby producing VFAs [33,34]. Huang et al. [10] found that the 6 mM gallic acid increased total gas and methane by 7% and 6%, respectively, after a 24 h incubation, while 2 mM and 4 mM gallic acid had no effect on gas production. However, Wei et al. [13] reported that 4.8 mM gallic acid promoted carbohydrate fermentation and alleviated methane production in 48 h batch-culture incubations. Our results showed a mitigating effect on methane production at lower doses of gallic acid; however, at higher doses, there was an unexpected increase in total gas and methane production.

### 4.2. Fermentation Parameters

Supplementation of gallic acid resulted in decreased pH and NH_3_-N and increased MCP. Ruminal NH_3_-N concentrations depend on protein degradation to amino acids. Part of the amino acids and dietary energy are used to synthesize VFA and MCP. It is known that the hydroxyl moieties of phenolic compounds bind to proteins, forming complexes that are more resistant to microbial degradation, thereby reducing ruminal protein degradation. The increased levels of gallic acid contributed to the linear decrease in ruminal ammonia N concentrations in this study, possibly due to the inhibition of proteolysis, peptidolysis, amino acid deamination, or an increase in microbial protein synthesis, or a combination of these factors, as supported by the reduced concentrations of isobutyrate and isovalerate. The reduction in VFA production at lower doses of GA might be related to decreased nutrient degradation and alterations in the composition of the ruminal microbiota. Lower doses of gallic acid might reduce the degradation of readily degradable substrates like protein and starch due to the inhibitory effect on amylolytic and proteolytic bacteria. Furthermore, gallic acid can be degraded to VFAs by ruminal microorganisms, suggesting that the increased VFA production at higher doses of GA may result from the further degradation of GA [13]. Gallic acid was reported to increase the acetate proportion by 23%. It can be degraded by rumen microbes into phloroglucinol, which is then transferred to produce acetate [10]. Getachew et al. [35] showed that adding 50 to 100 mg/g DM GA to alfalfa increased total VFA, acetate and butyrate production, and the A:P ratio after 24 h of incubation. Rira et al. [31] also reported that the increased acetate to propionate ratio could be due to the ruminal degradation of hydrolysable tannins by tannin-degrading bacteria, which convert them into acetate and butyrate. In this study, gallic acid played an important role in butyrate production, with butyrate formation providing an alternative pathway for metabolic hydrogen utilization compared to acetate. Phenolic compounds, particularly gallic acid, are considered suitable hydrogen acceptor alternatives and can be further converted into phloroglucinol or resorcinol by ruminal microbiota [36]. Additionally, it was reported that ruminal microbiota use hydrogen as electron donors to reduce phloroglucinol, producing acetate and butyrate [33]. Furthermore, the increased butyrate proportion may be due to the conversion of increased acetate into butyrate, as suggested by the role of phloroglucinol [37]. Butyrate production from acetate consumes metabolic hydrogen, potentially contributing to reduced hydrogen accumulation [10]. This could explain the observed increase in butyrate and decrease in CH_4_ production with the addition of 10 mg/g DM gallic acid in this study.

### 4.3. Microbial Diversity

Gallic acid tended to increase the observed species of rumen bacteria and reduce the Shannon and Simpson indices, suggesting that the gallic acid group had higher ruminal bacterial abundance but lower diversity. A greater number of bacterial species in the gallic acid group tended to have higher microbiome richness than the control. In our previous study, we found that gallic acid could improve richness and community evenness of rumen bacteria in preweaning calves, indicating a beneficial impact on rumen fermentation and development [14]. In vitro supplementation with phloroglucinol, a metabolite of gallic acid, was reported to increase bacterial abundance and reduce archaeal abundance without affecting protozoal and fungal abundance in dairy cows, but it reduced the abundance of protozoa, archaea, and fungi in dairy goats, reflecting a toxic effect on those communities [8,10]. Geerkens et al. [38] showed an increase in protozoal numbers when adding GA at 166.7 mg/g hay in batch culture. Soder et al. [39] also observed increased protozoal counts when incubating a high level of condensed tannin legume in rumen fluid. The antibacterial potency of the phenolic group may contribute to the antimicrobial activities of gallic acid. Generally, Gram-positive bacteria are thought to be more susceptible to gallic acid than Gram-negative bacteria, due to the absence of a protective outer membrane surrounding the cell wall. Herein, ß-diversity values indicated a low degree of similarity in the bacterial communities between the control and gallic acid groups.

### 4.4. Microbial Community

*Bacteroidetes* and *Bacillota* are two dominant bacterial phyla commonly found in the rumen fluid. *Bacteroidetes* play an important role in degrading proteins and carbohydrates; most strains are hemicellulolytic, proteolytic, or amylolytic, capable of breaking down structural carbohydrates, non-structural carbohydrates, and non-fibrous polysaccharides [40]. *Actinomycetota* members are known as rumen microorganisms that degrade carbohydrates and protein, and most strains produce amylases, cellulases, xylanases, and proteolytic enzymes [41]. Our results indicated lower abundances of *Bacteroidetes* and *Actinomycetota* in the GA group. Additionally, the *Bacteroides* genus and certain species, including *Bacteroides*sp.*CAG:545*, *Bacteroides*sp.*CAG:709*, *Bacteroides*sp.*CAG:770*, and *Bacteroides_unclassified*, were also less abundant in the GA group. Various *Prevotella* species are key in protein degradation and propionic acid production through the breakdown of starch, hemicellulose, and protein. Consequently, the reduced abundances of *Prevotella*sp.*E15-22* and *Prevotella_lacticifex* in the GA group led to decreased rumen protein degradation and propionic acid production [42]. The phylum *Planctomycetota* and *Verrucomicrobiota* are recognized as key degraders of lignocellulosic substrates in the herbivore gut, encoding a high number of carbohydrate-active enzymes per Mb of genome [43,44]. In this study, the comparison at the phylum level showed a significant increase in *Planctomycetota* and *Verrucomicrobiota* in the GA group. The genera *Butyrivibrio* and *Oscillospiraceae_unclassified* are known as butyrate producers [45]. The inclusion of gallic acid in this study increased the abundances of *Butyrivibrio* and *Oscillospiraceae_unclassified* genera, as well as the species *Butyrivibrio*sp.*AE2032* and *Butyrivibrio*sp.*XPD2006*. The abundance of certain genera and species was higher in the GA group, and we observed a trend towards increased butyrate concentrations. Inhibited methanogens may redirect dissolved H_2_ towards other hydrogenotrophic bacteria, leading to their transient increase in response to H_2_ accumulation [1]. Melgar et al. [46] suggested that H_2_-producing bacteria regulate the amount of H_2_ released under inhibited methanogenesis. This process also led to a significant increase in the molar proportion of butyrate in the rumen [1]. The inclusion of GA decreased the *Alistipes* genus and species *Alistipes*sp.*CAG:435* and *Alistipes*sp.*CAG:514*; some studies indicated that *Alistipes*, being pathogenic, may be reduced by the antibacterial properties of GA [47]. *Aristaeella_hokkaidonensis* can utilize dextrin, pectin, starch, xylan, and salicin to produce ethanol, hydrogen, acetate, and lactate, while *Aristaeella lactis*, isolated from bovine rumen, is related to significant lactate production [48]. Our results indicated an increase in the abundances of *Aristaeella_lactis* and *Aristaeella_hokkaidonensis* species in the GA group. *Ruminococcus flavefaciens*, a cellulolytic and hydrogen-producing bacterium essential for rumen fiber digestion, showed a decrease in the GA group [49]. The treatment of rice straw with liquid hot water has also been reported to reduce 16S rRNA gene copies of *Ruminococcus flavefaciens* and decrease CH_4_ emission [50].

Archaea utilize H_2_ to produce CH_4_ in the rumen, thereby maintaining a lower H_2_ partial pressure. *Euryarchaeota* was the dominant archaeal phylum, with *Methanobacter* being the predominant genus. At both the phylum and genus levels, a decrease in *Candidatus_Heimdallarchaeota* was observed in the GA group. At the species level, GA reduced the relative abundances of *Methanobrevibacter_thaueri*, *Methanobrevibacter_boviskoreani*, *Methanobrevibactersp.AbM4*, and *Methanosphaerasp.Vir-13MRS*. In this study, GA addition may reduce CH_4_ emission by altering the archaeal community composition by decreasing the abundance of several species belonging to the *Methanobrevibacter* genus, such as *Methanobrevibacter_thaueri*, *Methanobrevibacter_boviskoreani*, and *Methanobrevibactersp.AbM4*. Species of the genus *Methanobrevibacter* or members of the *Methanobacteriaceae* family were dominant methanogens in the rumen, growing and producing methane from H_2_ and CO_2_ [51,52]. *Methanobrevibacter*sp.*Abm4* is both a hydrogentrophic and methyltrophic methanogen that synthesizes coenzyme M, catalyzing the final step of all CH_4_ production reactions. GA reduced the abundance of *Methanobrevibactersp.Abm4* upon inhibition of CH_4_ production. A lower relative abundance of methanogenic archaea was observed in the GA group, possibly because gallic acid reduced the competition of methanogens for hydrogen during fermentation, allowing hydrogen to be utilized by other processes. Tannins have also been reported to inhibit the growth of methanogens, which are the primary agents of methanogenesis in the rumen; hydrolysable and condensed tannins may differentially affect methanogen activity [28]. The inhibition of CH_4_ by lower doses of tannins has been suggested to result from their antimethanogenic and antiprotozoal activities, which may lead to a decrease in methanogenic archaea counts or a combined effect of reduced archaea and protozoa [25].

### 4.5. Microbial Functions in KEGG

The functional genes of the rumen microbiome identified by metagenomics offer a means to assess the functions of rumen microorganisms. The differentially abundant KEGG pathways, primarily associated with carbohydrate metabolism, include those involved in carbohydrate degradation and volatile fatty acid (VFA) biosynthesis, such as amino sugar and nucleotide sugar metabolism, starch and sucrose metabolism, glycolysis/gluconeogenesis, and pyruvate metabolism. These pathways are more adept at degrading complex substrates and producing rumen VFAs, which are the main energy sources. KEGG functions related to carbohydrate metabolism were enriched in the GA group, possibly involving pathways in pyruvate, propionate, and butyrate metabolism, which were found to be increased in this group. These results indicated that the microbiome in the GA group has a greater capacity to degrade carbohydrates into pyruvate, which is then preferentially used for propionate and butyrate production over methane and acetate biosynthesis. Collectively, these findings highlight the specific functional potential of the microbiome related to carbohydrate degradation and VFA biosynthesis, enhancing our understanding of the functional roles of the rumen microbiome in rumen fermentation and methane production.

## 5. Conclusions

This study suggested that a 10 mg/g DM dose of GA reduced DM and CP digestibility and mitigated methane emission; a 10 mg/g DM dose supplementation with GA decreased NH_3_-N, total VFA, acetate, and propionate concentrations but increased MCP and butyrate concentrations. The metagenomics results revealed that the impact of GA on ruminal methanogenesis and fermentation patterns is likely due to its effects on the community and functions of rumen microbiomes, particularly, bacterial and archaeal communities, both involved in carbohydrate metabolism in ruminants. In conclusion, the addition of GA might be a significant strategy for suppressing methane emission in the rumen. An optimal GA feeding rate in this in vitro study appeared to be 10 mg/g DM for dairy cows. However, further research is needed to verify the in vivo effect of GA over a longer period.

## Figures and Tables

**Figure 1 animals-15-01959-f001:**
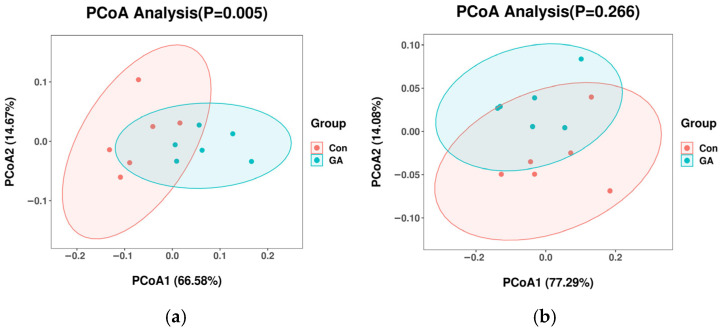
Principal coordinate analysis (PCoA) of rumen microbial communities: (**a**) bacteria; (**b**) archaea; (**c**) protozoa; (**d**) fungi. A(Con) = control group (0 mg/g DM, GA); B(GA) = gallic acid group (10 mg/g DM, GA).

**Figure 2 animals-15-01959-f002:**
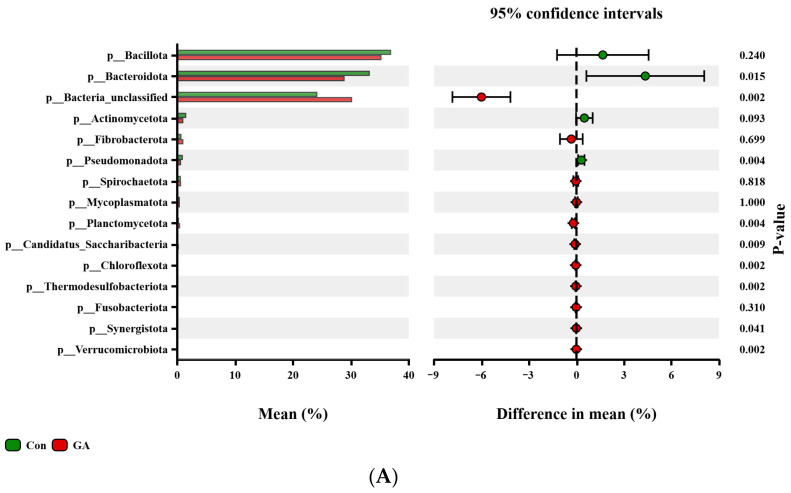
Comparison of the predominant bacterial community between control and gallic acid groups at the (**A**) phylum level, (**B**) genus level, and (**C**) significantly different species level. Con = control group (0 mg/g DM, GA); GA = gallic acid group (10 mg/g DM, GA).

**Figure 3 animals-15-01959-f003:**
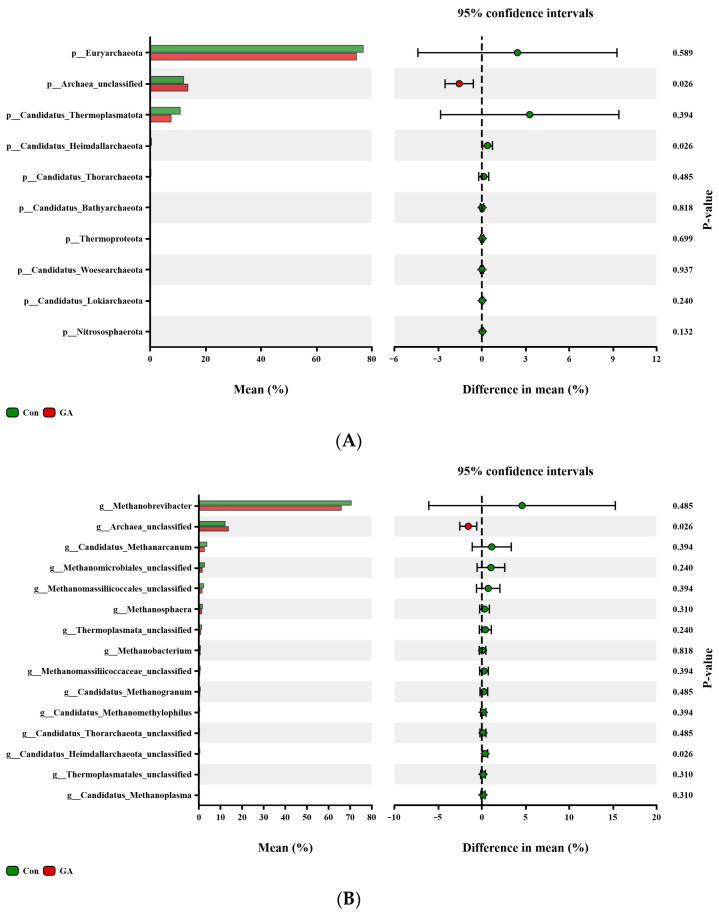
Comparison of the predominant archaeal community between control and gallic acid groups at the (**A**) phylum level, (**B**) genus level, and (**C**) significantly different species level. Con = control group (0 mg/g DM, GA); GA = gallic acid group (10 mg/g DM, GA).

**Figure 4 animals-15-01959-f004:**
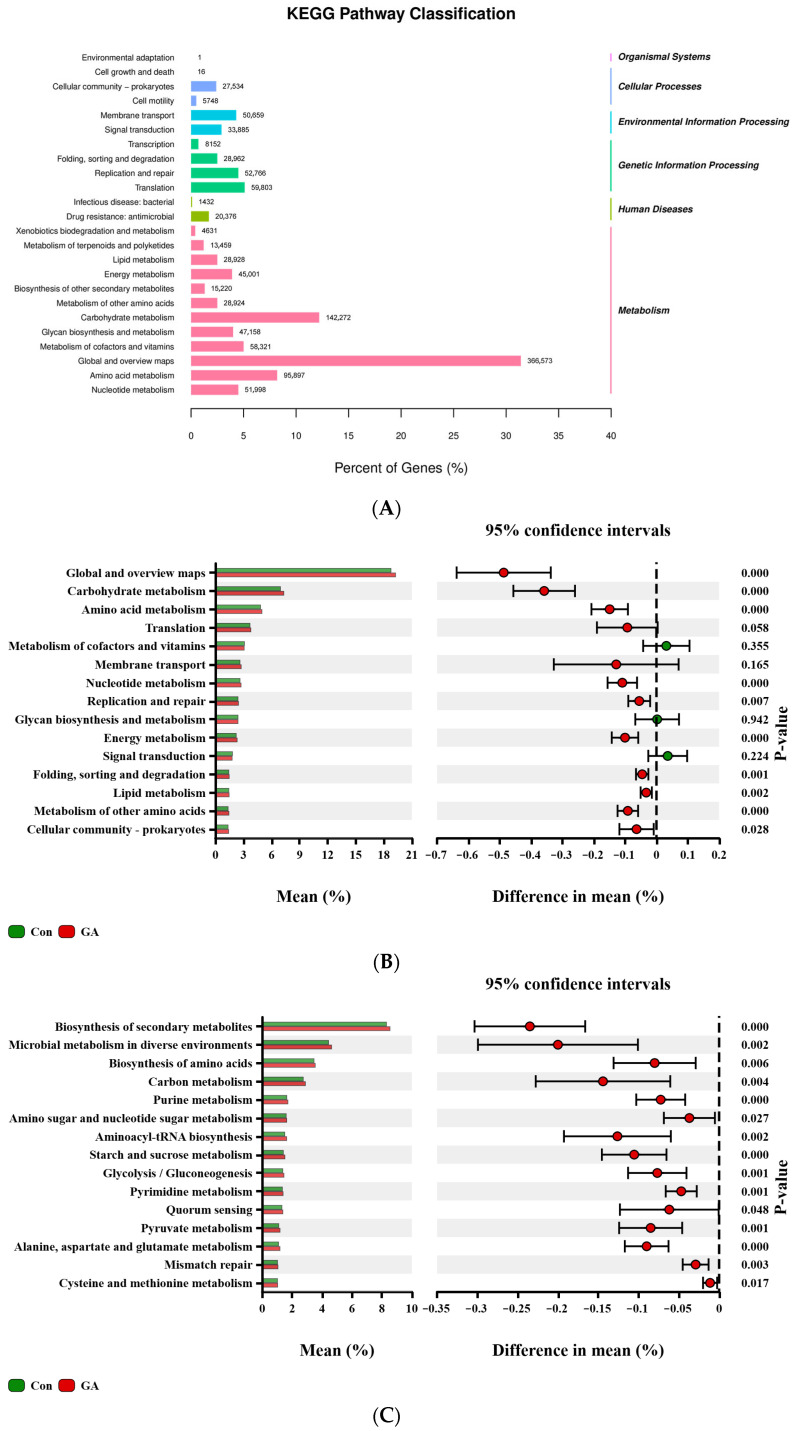
Significantly different (**A**) KEGG classifications; (**B**) KEGG functions at level 2; and (**C**) significantly different KEGG pathway definitions at the bacterial level. Con = control group (0 mg/g DM, GA); GA = gallic acid group (10 mg/g DM, GA).

**Table 1 animals-15-01959-t001:** Ingredient and nutrient composition of the experimental ration (DM basis).

Ingredient ^1^	%	Nutrient Level ^2^	%
Corn silage	31.7	NE_L_ (Mcal/Kg)	1.8
Alfalfa hay	17.8	CP	18
Soybean meal	11.1	NDF	38.3
Rapeseed meal	2.3	ADF	25.8
Cottonseed meal	2.2	Ca	0.66
Steam flaked corn	18.7	P	0.36
Dried distillers’ grains with solubles	4.9		
Wheat bran	9.6		
Minerals and vitamins premix	1.7		
Total	100		

^1^ The premix provided the following per kg of diet: vitamin A 220,000 IU, vitamin D3 72,000 IU, vitamin E 2000 IU, D-biotin 40.0 mg, nicotinic acid amide 2000 mg, Mn (as manganese sulfate) 710 mg, Zn (as zinc sulfate) 2005 mg, Fe (as ferrous sulfate) 830.0 mg, Cu (as copper sulfate) 680.0 mg, and Co (as cobalt sulfate) 12 mg. ^2^ NE_L_ = net energy for lactation.

**Table 2 animals-15-01959-t002:** Effects of gallic acid on nutrient digestibility and gas production in vitro.

Item	GA Levels (mg/g of DM)					SEM	*p*-Value ^2^		
	Con	5	10	20	40		L	Q	C
Nutrient digestibility ^1^									
DMD, %	56.1 ^ab^	53.7 ^b^	52.2 ^b^	59.8 ^a^	60.2 ^a^	1.75	0.22	0.006	0.29
CPD, %	56.6 ^ab^	53.4 ^bc^	50.8 ^c^	55.5 ^ab^	59.5 ^a^	1.73	0.46	0.02	0.39
NDFD, %	29.8	29.4	30.3	30.3	29.7	0.92	0.57	0.79	0.61
ADFD, %	20.1	19.2	19.8	22.0	21.5	1.25	0.36	0.21	0.50
Gas production									
Total gas, mL	230.4 ^b^	230.1 ^b^	215.3 ^c^	229.6 ^b^	243.3 ^a^	3.69	0.30	0.05	0.01
CH_4_, mL	54.8 ^c^	49.4 ^d^	45.7 ^e^	58.9 ^b^	64.0 ^a^	1.09	0.09	<0.0001	0.003
CH_4_/total gas%	23.8 ^b^	21.5 ^c^	21.2 ^c^	25.7 ^a^	26.3 ^a^	0.33	0.0006	<0.0001	0.09
CO_2_, mL	149.2 ^b^	147.9 ^b^	135.4 ^c^	155.4 ^b^	166.8 ^a^	3.12	0.66	0.001	0.003
CO_2_/total gas%	64.7 ^b^	64.3 ^b^	62.9 ^b^	67.8 ^a^	68.6 ^a^	0.92	0.06	0.006	0.09

^a–e^ Means within a row with different superscripts differ (*p* ≤ 0.05). ^1^ DMD = apparent degradability of DM; NDFD = apparent degradability of NDF; ADFD = apparent degradability of ADF. ^2^ L = linear; Q = quadratic; C = cubic.

**Table 3 animals-15-01959-t003:** Effects of gallic acid on rumen fermentation parameters in vitro.

Item	GA Levels (mg/g of DM)					SEM	*p*-Value ^1^		
	Con	5	10	20	40		L	Q	C
pH	6.74	6.73	6.69	6.69	6.69	0.021	0.06	0.99	0.57
NH_3_-N (mg/dL)	32.61 ^a^	29.39 ^b^	29.30 ^b^	28.36 ^b^	27.79 ^b^	1.066	0.01	0.29	0.41
MCP (mg/mL)	29.43 ^b^	30.61 ^b^	39.77 ^a^	39.48 ^a^	38.64 ^a^	2.165	0.0003	0.74	0.08
Total VFA (mM)	79.49 ^b^	75.29 ^b^	74.14 ^b^	85.16 ^a^	87.02 ^a^	1.961	0.08	0.0004	0.31
Concentration (mol/100 mol)									
Acetate	56.54 ^b^	56.44 ^b^	55.82 ^b^	58.24 ^a^	58.85 ^a^	0.540	0.07	0.03	0.15
Propionate	24.14 ^a^	23.26 ^ab^	23.43 ^ab^	22.91 ^b^	23.19 ^ab^	0.341	0.03	0.61	0.26
Butyrate	12.51 ^bc^	13.69 ^b^	14.96 ^a^	12.93 ^bc^	11.68 ^c^	0.427	0.20	0.0006	0.08
Isobutyrate	2.68	2.79	2.14	2.20	2.31	0.266	0.09	0.93	0.23
Valerate	2.29 ^ab^	2.00 ^bc^	1.85 ^c^	2.13 ^abc^	2.45 ^a^	0.125	0.27	0.03	0.58
Isovalerate	1.84 ^a^	1.83 ^a^	1.80 ^a^	1.59 ^ab^	1.52 ^b^	0.086	0.06	0.27	0.70
A/P	2.34 ^b^	2.44 ^ab^	2.40 ^ab^	2.54 ^b^	2.54 ^b^	0.054	0.03	0.63	0.19

^a–c^ Means within a row with different superscripts differ (*p* ≤ 0.05). NH_3_–N = ammonia nitrogen; VFA = volatile fatty acid; A/P = acetate to propionate ratio. ^1^ L = linear; Q = quadratic; C = cubic.

**Table 4 animals-15-01959-t004:** Effects of gallic acid on α-diversity indices of rumen microbial community in vitro.

Item	Con	GA	SEM	*p*-Value
Bacteria				
Observed species	15,793.83	16,195.83	133.821	0.06
Chao1	16,196.87	16,475.24	127.563	0.15
Shannon	7.94	7.61	0.019	<0.01
Simpson	0.956	0.922	0.0035	<0.01
Archaea				
Observed species	353.83	344.83	8.968	0.49
Chao1	370.84	370.55	5.067	0.97
Shannon	4.238	3.900	0.2252	0.31
Simpson	0.886	0.865	0.0156	0.37
Protozoa				
Observed species	88.33	95.50	1.755	0.02
Chao1	94.82	101.42	3.004	0.15
Shannon	4.29	4.33	0.048	0.54
Simpson	0.909	0.909	0.0025	0.99
Fungi				
Observed species	1230.33	1246.33	6.548	0.11
Chao1	1267.89	1278.08	9.80	0.48
Shannon	2.513	2.419	0.0207	0.009
Simpson	0.617	0.630	0.0044	0.054

## Data Availability

The original contributions presented in this study are included in the article. Further inquiries can be directed to the corresponding author.

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
