# Peer review of "Effects of Gallic Acid on In Vitro Ruminal Fermentation, Methane Emission, Microbial Composition, and Metabolic Functions"

_animals, 2025, doi:10.3390/ani15131959_

Round 1

Reviewer 1 Report

Comments and Suggestions for Authors

My comments and suggestions will be found in the original manuscript draft. 

Author Response

Reviewer: 1

C: It miss the summary finding of your research.

A: Thank you for pointing out the problem. We have now revised the manuscript and added the summary findings as suggested.

C: In the summary, you report only the objective and any finding is reported here. You can reduce the general information up and add two sentences related to your findings.

A: Thank you for pointing out the problem. We have streamlined the summary by reducing general background information and added key findings. We have delete “As a main greenhouse gas, methane absorbs atmospheric infrared radiation, contributing to climate change and global warming.”

C: Your abstract is very long (35 lines) more less the same as introduction (40 lines). As mentioned up, you have to improve the summary adding some finding there. Here in the abstract, be sure to put the key findings and not all of the study.

A: Thank you for pointing out the problem. According to your suggestion, We have made revisions to the summary. We have delete “Increasing GA doses linearly reduced the concentration of NH3-N (P = 0.01) and increased the concentration of microbial protein (MCP) (P = 0.0003)” “According to KEGG pathway definition, pathways in the category “carbohydrate metabolism” “within “Amino acid metabolism”.

C: Not clear, It was not completely randomized design. Suggestion: In vitro experiment was conducted testing GA dose levels (0, 5, 10, 20, and 40 mg/g DM) in cow's diet.

A: Thank you for pointing out the problem. According to your suggestion, we have revised “Experiment was conducted in the in vitro incubation by a completely randomized design and five culture GA dose levels (0,5,10, 20, and 40 mg/g DM) were tested in the trial” to “ In vitro experiment was conducted testing GA dose levels (0, 5, 10, 20, and 40 mg/g DM) in cow's diet”

C: 0 (g/g )DM.

A: Thank you for your suggestion! Thank you for pointing out the problem. According to your suggestion, we have revised “0 (g/L )DM to “0 (g/g )DM”

C: acetate to propionate ratio.

A: Thank you for pointing out the problem. According to your suggestion, we have revised “ratio of acetate to propionate” to “acetate to propionate ratio”

C: the species should be in italic.

A: Thank you for pointing out the problem. I'll carefully review the document and make the necessary italicization adjustments promptly to ensure the formatting meets the requirements.

C: In my opinion, this should be moved to the summary section.

A: Thank you for pointing out the problem. According to your suggestion, I have moved these contents to the summary section.

C: "in vitro" in italic.

A: Thank you for your suggestion! I'll carefully review the document and make the necessary italicization adjustments promptly to ensure the formatting meets the requirements.

C: Diet formulation and its chemical composition have to be reported in the table. It will be the table 1.

A: Thank you for your suggestion! According to your suggestion, The dietary formula and its chemical components have been reported in the table. It will be Table 1.

C: check the meaning of this sentence.

A: Thank you for your suggestion! According to your suggestion, We checked the sentence again. The revised version is as follows:The substrate was identical to the TMR fed to the donor cows. It was dried at 55°C for 48 h and then passed through a 1-mm sieve using a Wiley mill.

C: to stop the fermentation.

A: Thank you for pointing out the problem. According to your suggestion, we have revised “ terminate” to “ to stop the fermentation”.

C: You cannot determine the right DMD if you dried the nylon bags at 55°C.The normal temperature is 103°C. I know that drying the undigested materials at 103°C will modify the chemical analysis that you made further. But you should also determine the right DM by drying a sampled undigested materials at 103°C to correct the DMD and others (CP, NDF and ADF) as well.Explain or correct results

A: In fact, we determined the chemical composition of the residues in nylon bags after drying them at 55℃ for 48 hours, followed by drying them at 103℃ to correct the DMD and others (CP, NDF and ADF).

C: Ri= run effect (i=1, 2, ....). Specify the nombers of variables for this effect. Tj= treatment effect (j=1, 2, ....). Specify the nombers of variables for this effect.

A: Thank you for pointing out the problem. According to your suggestion, we have revised “Ri= run effect、Tj= treatment effect” to “Ri = run (i=1,2,3), Tj = treatment(j=1,2,3,4,5)”.

C: as random effect.

A: Thank you for pointing out the problem. According to your suggestion, we have revised “a random effect” to “ as random effect”.

C: I suggesdt to replace digestibility by degradability gas it was measured in vitro.

A: Thank you for pointing out the problem. According to your suggestion, we have revised “digestibility” to “degradability”.

C: NDF and ADF digestibility are not reported in the Table 1.

A: Thank you for pointing out the problem. According to your suggestion, The NDF and ADF digestibility data have now been added to Table 2 in the revised manuscript.

C: Use dry matter degradability to be consistent. Also use degradability for NDF and ADF.

A: Thank you for pointing out the problem. According to your suggestion, we have revised “apparent disappearance of DM、NDF、ADF” to “apparent degradability of DM、NDF、ADF”.

C: Suggestion: In vitro fermentation Parameters.

A: Thank you for your suggestion! According to your suggestion, we have revised “Fermentation Parameters” to “ In vitro fermentation Parameters” .

C: Suggestion: In vitro ruminal fermentation pH, NH3-N, MCP and VFA production of diet suplemented with gallic acid are presented in Table 2.

A: Thank you for pointing out the problem. According to your suggestion, we have revised “ Ruminal pH, NH3-N, MCP and fermentation parameters are presented in Table 2” to “In vitro ruminal fermentation pH, NH3-N, MCP and VFA production of diet suplemented with gallic acid are presented in Table 2”.

C: I am confused that the inclusion of 10 mg/g of DM exhibited the lowest total gas and CH4 (Table 1) and show here the highest propionate and butyrate concentration. Knowing that this volatile fatty acids are positively correlated with methane production. Could you have any explanation?

A: The production of propionic acid requires the consumption of hydrogen(Wang et al., 2023). In addition, gallic acid, as a phenolic compound, can be converted into phloroglucinol or catechol by rumen microorganisms as a hydrogen acceptor, and phloroglucinol will produce butyric acid under hydrogen reduction(Conradt et al., 2016; McSweeney et al., 2001). To sum up, the production of propionic acid and butyric acid requires hydrogen, and hydrogen is also a key substrate for methanogens to produce methane. Therefore, the 10mg/g dry matter inoculation group had the highest concentration of propionic acid and butyric acid, and its methane production was the lowest.

Conradt D, Hermann B, Gerhardt S, Einsle O, Müller M. Biocatalytic properties and structural analysis of phloroglucinol reductases. Angewandte Chemie International Edition 2016;55:15531-4

Mcsweeney C, Palmer B, Mcneill D, Krause D. Microbial interactions with tannins: Nutritional consequences for ruminants. Animal feed science and technology 2001;91:83-93

Wang K, Xiong B, Zhao X. Could propionate formation be used to reduce enteric methane emission in ruminants? Science of the Total Environment 2023;855:158867

C: Add L, Q and C in foottable.

A: Thank you for your suggestion! We have now added the abbreviations L (linear), Q (quadratic), and C (cubic) to the footnote of Table 3 as suggested. The revised version is included in the manuscript.

C: Acetate to propionate ratio.

A: Thank you for pointing out the problem. According to your suggestion, we have revised “Acetate/Propionate” to “Acetate to propionate ratio”.

C: tended to be higher.

A: Thank you for your careful review and helpful suggestion. According to your suggestion, we have revised “tended to higher” to “tended to be higher”.

C: Here, did you pool togheter all the gallic acid groups?

A: Thank you for pointing out the problem. We did not combine gallic acid groups together. We used a group of 10mg/g.

C: Figure 1 (a)、Figure 1 (b) and Figure 1 (b) and (c).

A: Thank you for pointing out the problem. I'll review the entire document and standardize the format, ensuring consistency throughout.

C: Decide to use one uppercase or lowercase.

A: Thank you for pointing out the problem. I'll review the entire document and standardize the format, ensuring consistency throughout.

C: decreased of.

A: Thank you for pointing out the problem. According to your suggestion, we have revised “decreased” to “decreased of”.

C: probably or primarily? Because you did measured separatly, right? It is an other raison to add the chemical composition of the diet and an other side add NDF and ADF degradability in the table 1.

A: Thank you for pointing out the problem. According to your suggestion, we have revised “primarily” to “probably”.

C: replace with: might be attributed ...

A: Thank you for pointing out the problem. According to your suggestion, we have revised “is” to “might be attributed”.

C: After this sentence add another one explaining that undigested protein in the rumen (protein by-pass) will be absorbed in the small intestine (Add reference).

A: Thank you for pointing out the problem. According to your suggestion, we have add “undigested protein in the rumen will be absorbed in the small intestine”.

JAVIER G, RABIAA M, ALBERTO G-G J, et al. Effects of correcting in situ ruminal microbial colonization of feed particles on the relationship between ruminally undegraded and intestinally digested crude protein in concentrate feeds [J]. Journal of the science of food and agriculture, 2018, 98(3): 891-5

C: considered as.

A: Thank you for your careful review and helpful suggestion. According to your suggestion, we have revised “considered” to “considered as”.

C: I suggest: on protein degradation to amino acid. A part of amino acid and provided diet emergy are used to syntheze VFA and MCP.

A: Thank you for pointing out the problem. According to your suggestion, we have revised “Ruminal NH3-N concentration depends on degradation of amino acid and their utilization for synthesis of MCP” to “Ruminal NH3-N concentration depends on protein degradation to amino acid. A part of amino acid and provided diet emergy are used to syntheze VFA and MCP”

C: delete are.

A: Thank you for your suggestion! According to your suggestion, We have deleted "are " .

C: check this sentence, I think hemicellulolytic break the structural carbohydrates.

A: Thank you for pointing out the problem. According to your suggestion, we have revised “ most strains are hemicellulolytic, proteolytic, or amylolytic, capable of breaking down non-structural carbohydrates and non-fibrous polysaccharides” to “most strains are hemicellulolytic, proteolytic, or amylolytic, capable of breaking down structural carbohydrates , non-structural carbohydrates and non-fibrous polysaccharides

C: Please this figure should be in results section and not discussion.

A: Thank you for your suggestion! According to your suggestion, The pictures in the discussion section have been adjusted.

C: It is not true for the VFA beacause the highest were observed for 20 and 40 mg of GA. Or you can precide saying again at 10 mg/g DM.

A: Thank you for pointing out the problem. According to your suggestion, we have revised “DM dose supplementation with GA decreased NH3-N, total VFA” to “a 10 mg/g DM dose supplementation with GA decreased NH3-N, total VFA.

C: bacterial and archaeal communities both involved in carbohydrate metabolism in ruminats.

A: Thank you for pointing out the problem. According to your suggestion, we have revised “bacterial and archaeal communities involved in carbohydrate metabolism” to “bacterial and archaeal communities both involved in carbohydrate metabolism in ruminats”.

C: Put italic "in vitro" in all the manuscript.

A: Thank you for your suggestion! I'll carefully review the document and make the necessary italicization adjustments promptly to ensure the formatting meets the requirements.

Reviewer 2 Report

Comments and Suggestions for Authors

The manuscript analyses the effects of gallic acid on in vitro ruminal fermentation, methane emission and microbial composition and metabolic functions. The overall data give valuable insight into this area. But there are some points that need to be improved as follows.

  1. Line 25 Delete "its function in dairy cows"
  2. Line 28 DM was first mentioned in the main context, should provide full name.
  3. Line 62 63 Use could potential.  
  4. Line 63 Delete dairy cows.
  5. Line 78 Methane production is a phenomenon in rumen metabolism and not a energetic inefficiency.
  6. Line 102 103 Gallic acid, as a hydrolysable tannin metabolite. Tannic acid is as a hydrolysable tannin.
  7. Line 107 Gallic acid and its metabolites.
  8. Line 110 Why 72-hour in vitro cultures was not used?
  9. Line 135 136 Was rumen fluid collected from stomach tube or fistulated animals? Add to the manuscript.
  10. Line 177 should be OTU not OUT.
  11. Line 177-186 OTU and QIIME are used in 16S rRNA analysis not metagenomic sequences.
  12. Line 200 CP, NDF and ADF was first mentioned in the main context, should provide full name.
  13. Line 304 Your study is on hydrolysable tannin and sources related to hydrolysable tannins should be reported.
  14. Line 340 Delete and.
  15. Double check again for clarity in writing and correct grammar.

Author Response

Reviewer: 2

C: Line 25 Delete “its function in dairy cows”

A: Thank you for your suggestion! According to your suggestion, We have deleted “ its function in dairy cows” .

C: Line 28 DM was first mentioned in the main context, should provide full name.

A: Thank you for your correction! We have added the full name of "DM" ("Dry Matter ") in Line 28. We will pay attention to the specification of first referencing terms in the future.

C: Line 62 63 Use could potential.

A: Thank you for pointing out the problem. According to your suggestion, we have revised “could potential” to “Use could potential”.

C: Line 63 Delete dairy cows.

A: Thank you for your suggestion! According to your suggestion, We have deleted “dairy cows”.

C: Line 78 Methane production is a phenomenon in rumen metabolism and not a energetic inefficiency.

A:

A: Thank you for pointing out the problem. According to your suggestion, we have revised “energetic inefficiency” to “phenomenon”.

C: Line 102 103 Gallic acid, as a hydrolysable tannin metabolite. Tannic acid is as a hydrolysable tannin.

A: Thank you for pointing out the problem. According to your suggestion, we have revised “Gallic acid, as a hydrolysable tannin metabolite” to “Tannic acid is as a hydrolysable tannin”.

C: Line 107 Gallic acid and its metabolites.

A: Thank you for your suggestion! I've made the change as you suggested.

C: Line 110 Why 72-hour in vitro cultures was not used?

A: Thank you for your insightful question regarding the 72-hour in vitro culture duration in our study. Based on previous research and methodological considerations, we selected this timeframe for the following reasons:

Reference 1: K. W, X.M. N, Y.G. Z, et al. Effects of propylene glycol on in vitro ruminal fermentation, methanogenesis, and microbial community structure [J]. Journal of Dairy Science, 2021, 104(3): 2924-34.

Reference 2: SCIENCE C O A, TECHNOLOGY G A U, NO. 1 YINGMEN VILLAGE ANNING, LANZHOU, GANSU, PEOPLE'S REPUBLIC OF CHINA, 730070, SCIENCE C O A, et al. Effects of oregano essential oil on in vitro ruminal fermentation, methane production, and ruminal microbial community [J]. Journal of Dairy Science, 2020, 103(3): 2303-14.

Reference 3: RUI L, YUEYU S, HAOKAI M, et al. Silibinin reduces in vitro methane production by regulating the rumen microbiome and metabolites [J]. Frontiers in Microbiology, 2023, 14: 1225643-.

C: Line 135 136 Was rumen fluid collected from stomach tube or fistulated animals? Add to the manuscript.

A: Thank you for your comment. We confirm that the rumen fluid samples were collected from fistulated animals. This detail has been added to the Materials and Methods section.

C: Line 177 should be OTU not OUT.

A: Thank you for pointing out the problem. According to your suggestion, the description related to OUT in the metagenomic determination method has been deleted.

C: Line 177-186 OTU and QIIME are used in 16S rRNA analysis not metagenomic sequences.

A: Thank you for pointing out the problem. The α-diversity index was calculated from the relative abundance data of microbial species using R software (version 4.3.2).

C: Line 200 CP, NDF and ADF was first mentioned in the main context, should provide full name.

A: Thank you for pointing out the problem. We have added the full name of CP (Crude Protein)、NDF( Neutral Detergent Fiber)、ADF(Acid Detergent Fiber). The revisions have been made in the manuscript, We will pay attention to the specification of first referencing terms in the future.

C: Line 304 Your study is on hydrolysable tannin and sources related to hydrolysable tannins should be reported.

A: Thank you for pointing out the problem. According to your suggestion, We report the sources related to hydrolysable tannins.

C: Line 340 Delete and.

A: Thank you for pointing out the problem. According to your suggestion, We have deleted “ and”.

C: Double check again for clarity in writing and correct grammar.

A: Thank you for pointing out the problem. We have thoroughly reviewed the text and implemented comprehensive revisions to address these concerns.